# CircANKRD12 Is Induced in Endothelial Cell Response to Oxidative Stress

**DOI:** 10.3390/cells11223546

**Published:** 2022-11-09

**Authors:** Christine Voellenkle, Paola Fuschi, Martina Mutoli, Matteo Carrara, Paolo Righini, Giovanni Nano, Carlo Gaetano, Fabio Martelli

**Affiliations:** 1Molecular Cardiology Laboratory, IRCCS Policlinico San Donato, San Donato Milanese, 20097 Milan, Italy; 2Operative Unit of Vascular & Endovascular Surgery, IRCCS Policlinico San Donato, San Donato Milanese, 20097 Milan, Italy; 3Department of Biomedical Sciences for Health, University of Milan, 20133 Milan, Italy; 4Laboratorio di Epigenetica, Istituti Clinici Scientifici Maugeri IRCCS, 27100 Pavia, Italy

**Keywords:** endothelium, circular RNAs, oxidative stress, circRNA–miRNA–mRNA regulatory network, p53 signaling pathway, Foxo signaling pathway

## Abstract

Redox imbalance of the endothelial cells (ECs) plays a causative role in a variety of cardiovascular diseases. In order to better understand the molecular mechanisms of the endothelial response to oxidative stress, the involvement of circular RNAs (circRNAs) was investigated. CircRNAs are RNA species generated by a “back-splicing” event, which is the covalent linking of the 3′- and 5′-ends of exons. Bioinformatics analysis of the transcriptomic landscape of human ECs exposed to H_2_O_2_ allowed us to identify a subset of highly expressed circRNAs compared to their linear RNA counterparts, suggesting a potential biological relevance. Specifically, circular Ankyrin Repeat Domain 12 (circANKRD12), derived from the junction of exon 2 and exon 8 of the ANKRD12 gene (hsa_circ_0000826), was significantly induced in H_2_O_2_-treated ECs. Conversely, the linear RNA isoform of ANKRD12 was not modulated. An increased circular-to-linear ratio of ANKRD12 was also observed in cultured ECs exposed to hypoxia and in skeletal muscle biopsies of patients affected by critical limb ischemia (CLI), two conditions associated with redox imbalance and oxidative stress. The functional relevance of circANKRD12 was shown by the inhibition of EC formation of capillary-like structures upon silencing of the circular but not of the linear isoform of ANKRD12. Bioinformatics analysis of the circANKRD12–miRNA–mRNA regulatory network in H_2_O_2_-treated ECs identified the enrichment of the p53 and Foxo signaling pathways, both crucial in the cellular response to redox imbalance. In keeping with the antiproliferative action of the p53 pathway, circANKRD12 silencing inhibited EC proliferation. In conclusion, this study indicates circANKRD12 as an important player in ECs exposed to oxidative stress.

## 1. Introduction

Reactive oxygen species (ROS) are essential mediators of both physiological and physio-pathologic signaling in the endothelium. The effects of ROS on cellular function depend on various parameters, including the localization, concentration and exposure duration of the stimulus. Complex cellular stress response mechanisms have evolved to attenuate the detrimental effects of oxidative stress. Indeed, an imbalance in redox homeostasis can lead to oxidative stress, which has a causative role in many vascular diseases [1,2].

EC oxidative stress is triggered by oxidants produced from activated immune cells or as a result of increased production of ROS from intracellular enzymes such as uncoupled nitric oxide synthase 3 (eNOS) and nicotinamide adenine dinucleotide phosphate (NADPH) oxidase, as well as from mitochondrial respiration, in the absence of neutralizing antioxidant mechanisms.

From an experimental point of view, studying the responses activated by cell exposure to a single, chemically defined, oxidative stress stimulus, such as a H_2_O_2_ pulse, might be particularly helpful for our understanding of the molecular mechanisms behind more complex situations associated with redox imbalance, such as hypoxia, senescence and critical ischemia.

As redox imbalance elicits EC dysfunction, it also triggers specific redox-sensitive gene expression changes, both transcriptional and posttranscriptional [3,4,5,6]. Important hubs of cell response to ROS and oxidative stress are Forkhead Box O1 (FOXO1) and NFE2-like bZIP transcription factor 2 (NRF2), which activate the antioxidant defences, Nuclear Factor Kappa B Subunit 1 (NFkB), which induces the expression of pro-inflammatory cytokines, and p53, which coordinates a complex antiproliferative and pro-apoptotic transcriptional response [3,7].

While the role of protein-coding RNAs in the EC response to redox imbalance has been extensively investigated [3,8], the relevance of non-protein-coding RNAs has started to emerge more recently [5,9,10]. In particular, circular RNAs (circRNAs) are covalently closed-loop structure RNAs generated by back-splicing, in which a downstream 5′ splice site is ligated to an upstream 3′ splice site across an exon or exons [11,12].

For a long time, circRNAs were dismissed as rare aberrations of the splicing process, but it is now clear that circRNAs are an important class of regulatory RNAs in a wide range of physiological and pathological settings, including cardiovascular diseases [13,14,15].

While many circRNAs are likely incidental products of the RNA splicing process, some circRNAs are more abundant than their linear counterparts [16], a telltale sign of likely biological relevance. Indeed, specific biological functions have been attributed to a fast-growing number of circRNAs [13,14,15]. In particular, many circRNAs display sequences complementary to a specific microRNA (miRNA), sponging it out, and hence decreasing its bioavailability for the inhibition of target mRNA [17,18,19,20].

For instance, silencing of circular zinc finger protein 609 (circZNF609) improves the endothelial dysfunction associated with high glucose and hypoxia stress, at least in part, by sponging miR-615-5p activity, leading to increased Myocyte Enhancer Factor 2A (MEF2A) expression [21], and several circRNA–miRNA–mRNA networks associated with oxidized low-density lipoprotein endothelial injury and atherogenesis have been reported [22].

Thus, while increasing evidence of circRNA’s role in EC physiopathology is accumulating, further insight is needed to understand its role in the EC response to oxidative stress. In this study, we investigated the circRNA landscape of human ECs exposed to H_2_O_2_ to identify a subset of highly expressed circRNAs, and a combination of bioinformatics, molecular biology and functional studies was adopted for their functional characterization.

## 2. Materials and Methods

### 2.1. Bioinformatics Analysis of RNAseq Data

Previously published RNAseq data (GSE104664) [5], revealing the transcriptomic profiles of HUVEC exposed to H_2_O_2_ (16 and 36 h) or solvent alone (16 h), were reanalyzed for detection of circRNAs by applying CIRI2 (v. 2.0.6; Gao Y., Wang J., Zhang J. and Zhao F., Beijing, China) [23,24]. Unique back-splice events identified in at least one of the nine libraries were filtered for circRNAs with a minimum of 5 counts in at least three samples. For calculation of circular-to-linear ratios, we first quantified all annotated linear junctions involved with either the donor or the acceptor site of the back-splice event. Then, the normalized, averaged counts of the linear junction with the highest expression were divided by the normalized, averaged counts of the back-splice junction. Finally, genes displaying ratios of ≥0 in all three conditions were ranked decreasingly by the circular-to-linear ratio and the top 20 were chosen for validation by qPCR.

### 2.2. Cell Cultures and Transfection

HUVEC (Gibco) were cultured in EGM-2 (Lonza, Verviers, Belgium) and exposed to 200 µM H_2_O_2_ [5] or hypoxia (1% oxygen) [25], as described before. These conditions were optimized and characterized in previous studies. Specifically, the H_2_O_2_ concentration used induced a strong cytostatic effect and very low EC death in vitro [5]. Likewise, the conditions used for the hypoxia experiments showed the expected impact on the coding and non-coding transcriptome [25,26,27]. For the knockdown of either linear or circular ANKRD12, 50 nM of transcript-specific ON-TARGETplus siRNAs (Dharmacon, Horizon Discovery, Waterbeach, UK) were transfected in 60% confluent HUVEC using siRNA transfection reagent (Santa Cruz Biotechnology; Dallas, TX, USA) according to the manufacturer’s protocol. In addition, the ON-TARGETplus non-targeting siRNA #1 (Dharmacon, Lafayette, LA, USA) was used as negative control. The chosen siRNA concentration resulted in an efficient knockdown in previous studies [25,28,29,30] and was reconfirmed for the present investigation by preliminary experiments (data not shown). Sequences are shown in Appendix A.

### 2.3. RNA Extraction and RNase R digestion

Total RNA was isolated from cells or tissues by TRIzol reagent, following the manufacturer’s instructions (Thermo Fisher Scientific Inc., Waltham, MA, USA). The purity and integrity of the obtained RNAs was measured by NanoDrop One (Thermo Fisher Scientific Inc., Waltham, MA, USA) and 2100 Bioanalyzer (Agilent Technologies, Santa Clara, CA, USA).

RNase R digestion was performed by incubating 1 µg total RNA with 1 U RNase R (Epicentre Biotechnologies, Madison, WI, USA) in 1× RNase R buffer in a 20 µL reaction at 37 °C for 10 min, followed by heat inactivation at 95 °C for 3 min. Control samples were treated the same way but without adding of RNase R [31].

### 2.4. Real-Time Reverse Transcriptase qPCR

To measure the expression levels of all transcripts, total RNAs were reverse-transcribed using the GoScript Reverse Transcription System (Promega Corporation, Madison, WI, USA) and random hexamers for priming the first-strand cDNA synthesis. The obtained cDNAs were then investigated by SYBR green qPCR (GoTaq qPCR Master Mix, Promega Corporation) according to the manufacturer’s instructions on a StepOne Plus Instrument (Thermo Fisher Scientific Inc., Waltham, MA, USA). Next, the relative expression was calculated using the comparative Ct method 2^−∆∆CT^ [32], normalizing to the averaged Cts of Ribosomal Protein L23 (RPL23) and Ubiquitin C (UBC). Finally, the circular-to-linear ratio was estimated by subtracting the raw Ct of the linear transcript from the raw Ct of the corresponding circular transcript.

Primer couples were designed with the assistance of Primer-BLAST tool (Appendix A) [33]. The circRNAs primers spanned the back-splice junction, while the primer couples for the linear transcripts crossed the linear junction to a neighboring exon (Appendix A). The specificity of primers was assessed by performing melting curve analysis and verification of expected amplicon size by agarose gel electrophoresis. In addition, primer efficiency analysis was verified for circular and linear transcripts of ANKRD12.

### 2.5. Capillary-like Network Formation and Growth Curves

For capillary-like network formation assay, 48 h after transfection HUVEC were seeded at a density of 18,000 cells per chamber of an 8-well chamber slide coated with Matrigel (Cultrex Reduced Growth Factor Basement Membrane Extract, PathClear, Bio-Techne, Minneapolis, MN, USA). Four chamber slides were prepared for each transfection and four pictures for each chamber were taken, counted and averaged. The number of branches of each node as well as the number of tubes was counted by the blind.

For growth curves resulting from cell number count, HUVEC were seeded 24 h after transfection and trypan blue negative cells were measured by the Countess II cell counter at 0, 24 and 48 h after seeding (Thermo Fisher Scientific Inc., Waltham, MA, USA). For growth curves derived from monitoring the cell viability, HUVEC were first seeded in 96 wells and then transfected. Cells were washed at 0, 24, 48 and 72 h after transfection and a cell viability assay was performed using crystal violet staining as described before [34]. For both growth curve methods, the independent experiments were aggregated according to the following procedure: First, for each experiment, we expressed the obtained value for each replicate as a percentage relative to time point 0 (set at 100%). Then, the resulting percentages were averaged for each time point across all experiments.

### 2.6. Patient Characteristics and Tissue Collection

Skeletal muscle biopsies of 10 patients, seven females and three males, aged 76.4 ± 6.6, affected by CLI of the leg and undergoing above-the-knee amputation, were harvested at the amputation site. Altogether, 34 biopsies of different muscles were taken: tibialis, sartorius, gastrocnemius, popliteus, gemelli, surae or peroneal. As controls, we used ten non-ischemic biopsies of biceps, sartorius or vastus taken from 10 individuals, three females and seven males, aged 57.3 ± 13. Samples were treated as described before [5]. The IRCCS San Raffaele ethical committee approved this study, protocol code miRNA CLI, number 69/INT/2016 of 05/05/2016.

### 2.7. Identification of circRNA–miRNA–mRNA Network

miRNA binding sites for circANKRD12 were identified by two different web tools, circular RNA Interactome (update 01.30.2020; M. Gorospe’s Lab, Baltimore, MD, USA) [35] and ENCORI (accessed on 15 April 2022; RNA Information Center, Guangzhou, China) [36]. The microRNA Data Integration Portal mirDIP (version: 5.0.2.3.; Jurisica Lab, Toronto, ON, Canada) [37] was used to predict targets of these miRNAs, considering only the highest score class (i.e., top 1%). The obtained output was intersected with mRNAs found to be induced upon exposure to 16 or 36 h of H_2_O_2_ in HUVEC identified by our group previously [5]. Using the tool ShinyGO (version 0.75; Ge SX, Jung D and Yao R, Brookings, SD, USA) [38], we analyzed the enrichment for KEGG pathways and generated lollipop graphs, sorted by FDR.

### 2.8. Statistical Analysis

GraphPad Prism 7.01 (GraphPad Software Inc., San Diego, CA, USA) was used for statistical analysis and for graph generation. Data were checked for distribution by D’Agostino and Pearson normality testing, whenever relevant. We used a two-tailed student’s *t*-test, or Mann–Whitney, to identify statistically significant differences as appropriate. If not indicated otherwise, the threshold of significance was set at *p* < 0.05. Values are expressed as ±standard error.

## 3. Results

### 3.1. Identification of circRNA Expression in HUVEC under Oxidative Stress by RNA Sequencing

In order to define the circRNA landscape of ECs exposed to oxidative stress, we took advantage of a previously published RNA sequencing dataset, generated from rRNA-depleted RNAs (GSE104664, Fuschi 2017). The sequenced libraries were prepared from HUVEC exposed to a sublethal dose of H_2_O_2_ (200 µM) for 16 and 36 h in triplicate; HUVEC treated with solvent alone for 16 h were chosen as controls to avoid the possible confounding effects of cell confluency observed at 36 h.

Expressed circRNAs were identified by applying the CIRI2 algorithm [23,24] (Figure 1). Nearly 2500 unique back-splice events were found; of those, >350 showed a robust expression in three or more samples (Appendix A). However, despite several attempts to identify differential expression, none of the resulting candidates could be validated by qPCR. Therefore, we changed the approach and identified circRNAs with a similar or even higher expression than their linear counterparts, suggesting a potential biological relevance [16]. To discover these particularly interesting circRNAs, the circular-to-linear ratios were calculated by comparing the adjacent linear junction with the highest expression to the back-splice junction. The circular-to-linear ratios were then ranked decreasingly and the top 20 were chosen for investigation by qPCR of their possible regulation by oxidative stress (Appendix A).

### 3.2. Identification of Differentially Expressed circRNAs in HUVEC Exposed to Oxidative Stress by qPCR

Among the 20 circRNAs displaying potential relevance due to their circular-to-linear ratios, we were able to generate an efficient qPCR primer for both the circular and the linear isoform for 16 (Appendix A and Appendix A). An independent set of HUVEC with an experimental set-up identical to that adopted for the transcriptomic study was used to validate the results and to investigate a possible differential expression of linear and circular transcripts due to oxidative stress (Figure 2). All 16 circRNAs tested were readily detectable. For high-confidence validation, a stringent significance threshold for differential expression was applied (*p* < 0.01). ANKRD12 was unique in showing an induction of the circular-to-linear ratio robustly at both time points. Furthermore, the circular transcript displayed significant modulation at 16 h (Figure 2a). Three other candidates, the F-Box and WD Repeat Domain Containing 7 (FBXW7), Arginine–Glutamic Acid Dipeptide Repeats (RERE) and Solute Carrier Family 8 Member A1 (SLC8A1), displayed statistically significant differences for the circular-to-linear ratio only at one time point, indicating that the modulation was not sustained. Moreover, ADAM Metallopeptidase with Thrombospondin Type 1 Motif 6 (ADAMTS6) and Gon-4-like (GON4L) showed modulation only for the circular transcript at 16 h of treatment but not for the circular-to-linear ratios, suggesting that the whole genomic locus was modulated in the same direction.

Thus, circANKRD12 induction upon H_2_O_2_ treatment was further validated, using independent HUVEC cultures and a higher number of replicates. We could reconfirm the increase in circular-to-linear ratios at both time points (Figure 2b). Additionally, induction of the circular transcript was significant at the time point of 36 h (Figure 2c). Therefore, circANKRD12 was chosen for characterization and further investigation.

### 3.3. Characterization of circANKRD12 in Endothelial Cells

Due to the absence of free 5′ and 3′ ends, a significant characteristic of circRNAs consists of their higher resistance towards exonucleases compared to their linear counterparts [39,40]. Thus, as a first step for the characterization, we investigated the effect of RNase R digestion on the transcripts of ANKRD12, comparing the linear to the circular isoforms. Indeed, RNase R degrades all linear RNA forms with 3′ single-stranded regions longer than seven nucleotides [41]. We found that, while ca. 70% of the circular transcript resisted RNase R digestion, less than 2% of the linear transcript was detected by qPCR when comparing digested to non-digested transcript (Appendix A). Subsequently, Sanger sequencing was performed to analyze the transcriptional structure of circANKRD12 in HUVEC. The back-splice junction of the circANKRD12 transcript investigated in this study is located between exon 2 and 8 (hsa_circ_0000826). In a previous study on breast cancer, two isoforms of this circRNA were identified: a short transcript, consisting of only two exons (i.e., exons 2 and 8), and a long transcript, consisting of all exons between 2 and 8 but skipping exon 4 [42]. In ECs, Sanger sequencing with amplicons produced by various couples of divergent primer pairs reading across the back-splice junction showed that two different long isoforms of circANKRD12 were expressed. Both circANKRD12 isoforms spanned from exon 2 to exon 8, one retaining exon 4 (994 bp length) and one lacking exon 4 (925 bp length), (Figure 3).

Additionally, we wanted to verify if the short isoform observed in human cancer cells was also present in human ECs. For this purpose, we designed divergent primers located on exon 8, amplifying preferentially the short isoform of only 286 bp in length. The expression levels obtained by qPCR were then compared to the expression derived from back-splice spanning primers, amplifying any circular isoform of circANKRD12, showing a several 100-fold higher expression compared to the short isoform (Appendix A). Agarose gel analysis confirmed the expected size of the amplicon produced by the primers preferentially amplifying the short isoform (Appendix A). Hence, we conclude that the short circANKRD12 isoform is present in HUVEC but at negligible concentrations compared to the longer isoforms.

### 3.4. Increased Expression of circANKRD12 in HUVEC Exposed to Hypoxia

Next, we wanted to investigate the possible role of circANKRD12 in ECs exposed to hypoxia, another stress factor associated with mitochondrial dysfunction and increased oxidative stress [43,44,45]. To this end, we exposed HUVEC to 1% O_2_ for 24 and 48 h or to normoxia for 24 h. qPCR analysis showed significant induction of circANKRD12 upon hypoxia at both time points (Figure 4). Conversely, linear ANKRD12 expression did not change at any of the time points in a significant manner (Figure 4a). Accordingly, the circular-to-linear ratio was found to be increased at both time points (Figure 4b).

### 3.5. Increased Circular-to-Linear Ratio of ANKRD12 in CLI

CLI has been associated with increased hypoxia levels and oxidative stress in ischemic muscles [46,47,48]. Therefore, we investigated the expression levels of circular and linear ANKRD12 by qPCR in skeletal muscle samples taken from the ischemic limb of patients undergoing amputations for CLI. For this purpose, different muscles were harvested at the amputation site (tibialis, sartorius, gastrocnemius, popliteus, gemelli, surae or peroneal) and compared to non-ischemic biopsies taken from the biceps brachii, sartorius or vastus of control subjects. Analyzing linear and circular transcripts of ANKRD12 separately, we did not find a significant modulation in ischemic muscles (Figure 5a). However, estimating the circular-to-linear ratio revealed a significant induction, hinting at a possible relevance of the circular transcript in the ischemic tissues (Figure 5b). 

### 3.6. Knockdown of circANKRD12 Impairs the Formation of Capillary-like Structures

Oxidative stress plays an important role in angiogenesis [49] and one way to assay certain features of angiogenesis in vitro is to exploit the ability of ECs to form capillary-like structures on Matrigel [50].

In order to investigate the role of circular and linear ANKRD12 in forming capillary-like structures, using a loss-of-function approach, transcript-specific siRNAs were generated. Altogether, seven siRNAs, targeting specifically either the back-splice junction [42] or an exon outside of the circANKRD12 boundaries (exon 9, 11, 12), were screened (Appendix A). Comparison to a negative control (non-targeting siRNA) allowed the identification of siRNAs that either silenced the circular but not the linear isoform of ANKRD12 (si_Circ.1, targeting the splice junction) or, conversely, that knocked down linear but not circular ANKRD12 (si_Lin.3, targeting exon 9) (Appendix A).

Next, we first blocked the expression of either circular or linear ANKRD12 in HUVEC and then quantified the number of branch points (Figure 6a) and tubes (Figure 6b) formed after seeding on Matrigel (Figure 6c). This in vitro approach showed that only the knockdown of circANKRD12 significantly decreased the ability of HUVEC to form capillary-like structures, while, upon knockdown of linear ANKRD12, no significant effects could be observed.

### 3.7. Identification of the circRNA–miRNA–mRNA Network of ANKRD12 Reveals the Enrichment in Pathways Closely Related to Oxidative Stress

One fundamental mechanism of action of circRNAs is mediated by their ability to bind and sponge out specific miRNAs, therefore regulating the expression of their target mRNAs. Mounting evidence has indicated the dysregulation of the circRNA–miRNA–mRNA interaction network in the pathogenesis of many diseases [51], including oxidative stress-mediated pathologies [21,52,53]. To reveal potential interaction networks of circANKRD12 triggered by oxidative stress in ECs, we designed two pipelines using online prediction tools and the dataset of transcriptomic changes induced by H_2_O_2_ treatment in HUVEC cells we had previously published [5].

In the first step, miRNA binding sites for circANKRD12 were predicted by using CircInteractome [35] (Figure 7a) or ENCORI [36] (Figure 7b). Next, the potential targets of the identified miRNAs were predicted by mirDIP [37] (Figure 7). Although we obtained comparable numbers of predicted miRNAs, the miRNAs identified by the TargetScan-based CircInteractome had nearly twice as many predicted targets than the miRNAs selected by ENCORI, supported by Ago CLIP-seq experiments. Finally, to filter the mirDIP predictions for targets potentially relevant to the endothelial response to oxidative stress, we intersected them with the list of mRNAs significantly induced upon exposure to H_2_O_2_ [5], resulting in >200 and >100 mRNA targets within the CircInteractome- and the ENCORI-based pipelines, respectively. Remarkably, KEGG enrichment analysis of these targets identified the same two pathways as the most significantly enriched for both circANKRD12–miRNA–mRNA networks: Forkhead Box O (Foxo) and p53 signaling pathways (Figure 7 and Appendix A). This is particularly striking as both pathways have already been shown to play an essential role in the cellular response to redox imbalance [5,7,54].

In order to find experimental evidence of this bioinformatics analysis, we focused our attention on the p53 pathway. Specifically, given the p53 antiproliferative effect during cellular redox response [5,54,55], we investigated the potential impact of circANKRD12 on EC growth. To this aim, the expression of circANKRD12 was knocked down in HUVEC by transcript-specific siRNAs and cell growth was then monitored by either counting the cell numbers (Figure 8 and Appendix A) or measuring the crystal violet absorbance of viable cells (Appendix A). In both experiments, silencing of circANKRD12 resulted in a significant reduction in cell numbers, with an increasing effect over time.

## 4. Discussion

In this investigation, we studied the circRNA landscape of human ECs exposed to oxidative stress using an RNA sequencing approach. Differential expression analysis did not achieve a satisfying validation rate, likely due to the low number of reads spanning the junctions observed for most circRNAs. Instead, we used the circular-to-linear ratios to identify a set of candidates for further experimental investigation. This strategy is based on the suggestion that a high circular-to-linear ratio indicates an independent regulation of the circRNA species and, therefore, implies the potential for an additional biological function [16].

During the last years, an increasing number of circRNAs have been identified to play a role in diseases associated with oxidative stress [56], such as circular sodium/calcium exchanger 1 (circNCX1), mediating ischemic myocardial injury [57] or circular CDKN2B Antisense RNA 1 (circANRIL), reducing vascular endothelial injury, oxidative stress and inflammation in rats with coronary atherosclerosis [58]. Moreover, circSLC8A1 has been linked to oxidative stress-related Parkinsonism [59], whereas circSPECC1, transcribed from *Sperm Antigen with Calponin Homology and Coiled-Coil Domains 1* (SPECC1) is modulated upon H_2_O_2_ stimulation in hepatocellular carcinoma [60]. Among these circRNAs, circNCX1 and circANRIL were not expressed in our dataset, while circSPECC1 and circSLC8A1 were readily detectable but showed no robust modulation.

Here, we identified the induction of circANKRD12 in HUVEC during the response to oxidative stress triggered by H_2_O_2_. These findings are in accordance with recent studies, showing circANKRD12 to be highly expressed in various cancer cell types (ovarian, lung and breast cancer) [42] and in the peripheral blood of myocardial infarction patients [61]. In the latter study, the signature identified by RNAseq in infarcted patients was validated in a cardiomyocyte cell line (AC16) exposed to oxidative stress, confirming the induction of circANKRD12 [61]. However, in contrast to our findings, in AC16 cells, the expression of circSLC8A1 was positively modulated by intracellular oxidative stress, underlining the necessity to investigate each circRNA specifically in a relevant experimental system.

To verify the biological relevance of circANKRD12 in EC, first, we investigated the effects of its modulation on a prominent feature of ECs [49]: knockdown of circANKRD12, but not of linear ANKRD12, resulted in the impaired formation of capillary-like structures, suggesting a role of circANKRD12 in angiogenesis.

For a better understanding of the possible functions of circANKRD12, we identified circRNA–miRNA–mRNA networks using two different, carefully designed pipelines. Albeit starting from different miRNA prediction tools, showing only a partial overlap between their outputs, after filtering for experimental data, enrichment analysis eventually identified the same KEGG pathways as the most significant for both pipelines: p53 and FOXO pathways. This suggests that different prediction tools might be considered as complementary rather than less or more reliable. Indeed, both pathways have been identified as playing critical roles in the cell response to oxidative stress.

Transcription factors of the Foxo family are important regulators of the cellular stress response, promoting the cellular antioxidant defences and affecting cell death and proliferation [7]. For example, in vascular ECs, Foxo3a is a direct transcriptional regulator of a group of oxidative stress protection genes [62] and tube formation and migration were found to be inhibited by Foxo1 and Foxo3a [63].

P53 is activated by a myriad of stress stimuli, including reactive oxygen species, triggering a complex antiproliferative and pro-apoptotic transcriptional program [54,64]. Recently, our group highlighted the central role of the p53 pathway in the non-coding-RNA response to oxidative stress in human ECs. In detail, we found that upon H_2_O_2_ treatment, miR-192-5p was strongly induced by p53 and its overexpression significantly decreased EC proliferation, inducing cell death [5].

To validate the bioinformatics network analysis, we measured EC growth after silencing circANKRD12, highlighting the importance of circANKRD12 for EC proliferation. Thus, the identified reduced EC proliferation, together with the impaired formation of capillary-like structures [63], reveals a phenotype upon circANKRD12 inhibition corroborating experimentally the circRNA–miRNA–mRNA network we constructed in silico. In light of these observations, circANKRD12 induced in EC by oxidative stress might play a protective role by regulating cell proliferation and tubulogenesis.

Moreover, the circANKRD12–miRNA–mRNA network based on the ENCROI miR-prediction tool revealed senescence among the three most significantly enriched pathways. This is also particularly interesting since it has been shown that, in ECs, senescence can be induced by various insults, including oxidative stress [65,66,67].

Interestingly, the silencing of circANKRD12 in ovarian cancer cell lines resulted in the inhibition of cell proliferation and increased migration and motility [42]. While the observed reduction in cell growth is in keeping with our findings in ECs, the acceleration of migration and motility, both essential for angiogenesis [68,69], is in apparent contrast to the inhibited formation of capillary-like structures we found in ECs upon knockdown of ANKRD12. This discrepancy is likely due to cell-specific effects, confirming the need for testing each circRNA using the specific cellular background. In the same study, the authors investigated the effects of circANKRD12 knockdown on the mitochondrial oxygen consumption rate, finding a decreased oxidative phosphorylation in ovarian and breast cancer cell lines. Given that mitochondria are an important endogenous source of ROS [70], it is tempting to speculate that the relationship between circANKRD12 and oxidative phosphorylation observed in cancer may also be relevant for the EC response to oxidative stress.

Remarkably, ECs exposed to hypoxia, a different stress factor associated with mitochondrial dysfunction and increased oxidative stress [38,39,40], showed increased circANKRD12 levels, corroborating further a potential role of circANKRD12 during the response to redox imbalance.

CLI is associated with increased oxidative stress levels in the ischemic muscles and redox imbalance plays a causal role in ischemia-induced tissue damage [46,47,48,71]. Albeit circANKRD12 level alterations in the ischemic muscles of CLI patients were not significant, the circular-to-linear ratio was found to be significantly induced. This suggests that the identified involvement of circANKRD12 in the EC response to oxidative stress observed in vitro might also have relevance in physiopathological conditions in humans.

Although circRNAs are members of the non-coding RNAs, for a subset of them a protein coding activity has been identified [72,73]. We found that circANKRD12 contains a putative open reading frame, overlapping with the first 315 amino acids of the ANKRD12 mRNA product. Western blotting analysis performed with an antibody targeting the N-terminus of ANKRD12 failed to detect peptides potentially related to circANDRD12 translation (not shown). Still, we cannot rule out that circANRD12 translation occurs through internal ribosomal entry sites, resulting in an isoform not recognized by the commercially available antibodies [72,73].

In conclusion, RNAseq data derived from ECs exposed to H_2_O_2_ were exploited to discover circRNAs modulated by oxidative stress. Network analysis performed for the identified circANKRD12 indicated that p53 and Foxo pathways play a fundamental role in the oxidative stress response in many different systems. Accordingly, silencing of circANKRD12 affected known features of the redox imbalance response, i.e., proliferation and tubulogenesis, suggesting a potential role of circANKRD12 in the protection of ECs against oxidative stress.

## Figures and Tables

**Figure 1 cells-11-03546-f001:**
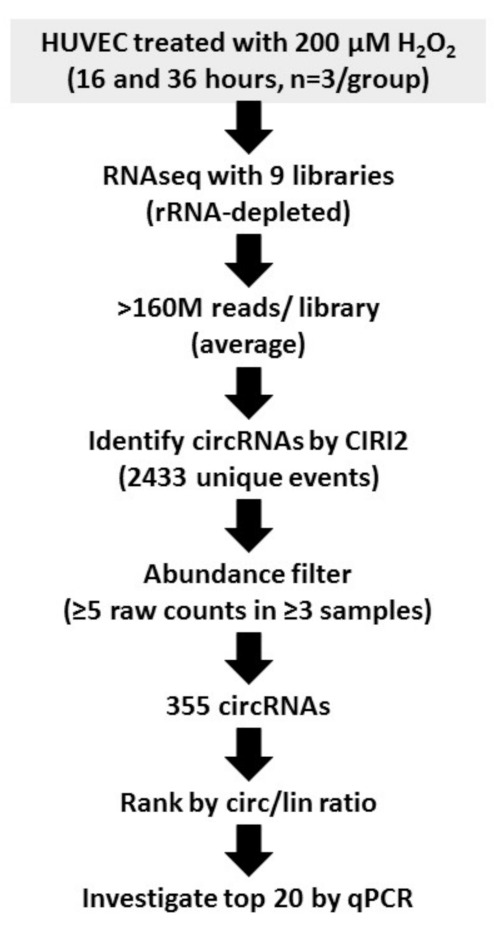
Bioinformatics pipeline for identification of circRNA expression in HUVEC exposed to oxidative stress. The experimental set-up of the previously published dataset (GSE104664) is shown together with applied software and subsequent filtering steps. HUVEC exposed to 16 h of solvent were used as controls.

**Figure 2 cells-11-03546-f002:**
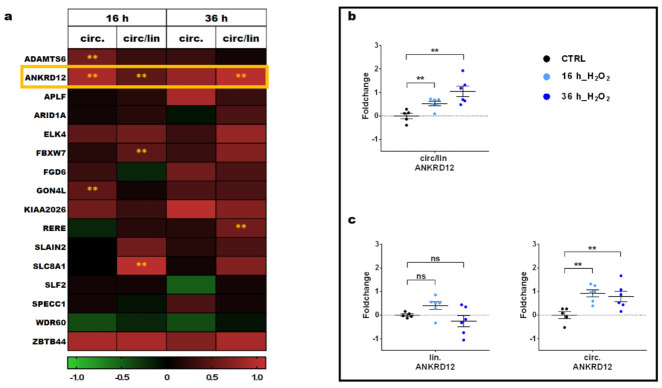
circANKRD12 is induced by H_2_O_2_ treatment of HUVEC. HUVEC were treated with 200 µM H_2_O_2_ for 16 and 36 h or with solvent alone for 16 h as control. Expression levels of linear and circular transcripts were measured by qPCR. All fold changes are in log2 scale (−ΔΔCT). Among all RNAs tested, only circANKRD12 displayed a sustained modulation. (**a**) Heatmap of technically conclusive candidates, showing differences in expression of H_2_O_2_-treated samples compared to controls as fold changes. The significance threshold was set at *p* < 0.01 (** *p* < 0.01; *n* = 3). (**b**) Scatterplot of circular-to-linear ratio (circ/lin) of ANKRD12 in HUVEC exposed to H_2_O_2_ compared to control (CTRL). (**c**) Scatterplots of the expression levels of circular (circ.) and linear (lin.) ANKRD12 transcripts in HUVEC exposed to oxidative stress compared to CTRL. For both panel (**b**,**c**), individual values are indicated by dots, while mean and SEM are indicated by horizontal lines (*n* = 6; ** *p* < 0.01; ns: not significant).

**Figure 3 cells-11-03546-f003:**
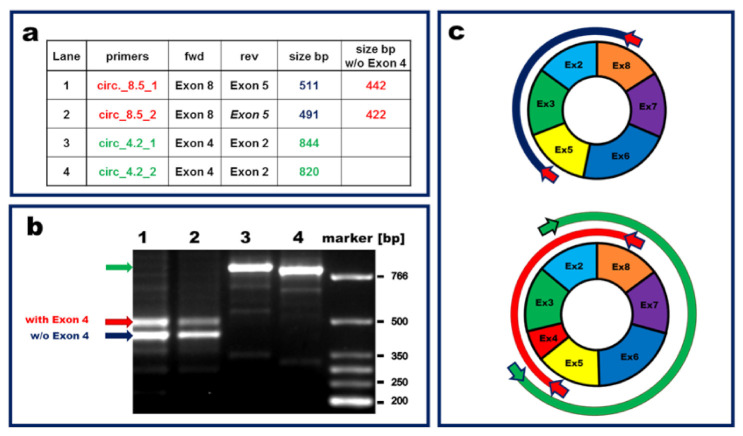
Characterization of circANKRD12 in human endothelial cells. Sanger sequencing was performed to explore the structure of circANKRD12 expressed in HUVEC. (**a**) Table showing details of divergent primer pairs designed to exclusively amplify circular transcripts sharing the back-splice junction between exon 2 and exon 8. (**b**) Agarose gel analysis performed with amplicons obtained by RT-PCR. Arrows indicate the bands extracted for sequencing. (**c**) Two circular isoforms of ANKRD12 as identified by Sanger sequencing. The blue band represents the sequenced amplicons derived from the circular transcript lacking exon 4. The green and red bands represent sequences derived from the transcript comprising exon 4. Arrows depict the primes used for RT-PCR.

**Figure 4 cells-11-03546-f004:**
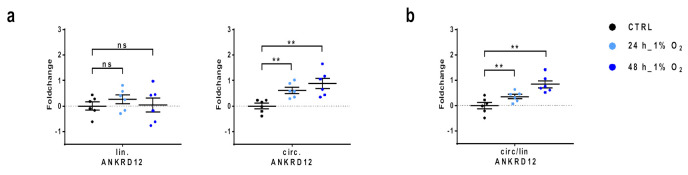
Induction of circular ANKRD12 in hypoxic HUVEC. HUVEC were exposed to 1% O_2_ for 24 and 48 h or to normoxia for 24 h (CTRL). Expression levels of the circular and linear transcripts were measured by qPCR. (**a**) Scatterplot of the expression levels of circular (circ.) and linear (lin.) ANKRD12 transcripts in hypoxic HUVEC compared to CTRL. (**b**) Scatterplot of the circular-to-linear ratio (circ/lin) of ANKRD12 in hypoxic HUVEC compared to CTRL. Fold changes are in log2 scale (−ΔΔCT) and individual values are indicated by dots, while mean and SEM are indicated by horizontal lines (*n* = 6; ** *p* < 0.01; ns: not significant).

**Figure 5 cells-11-03546-f005:**
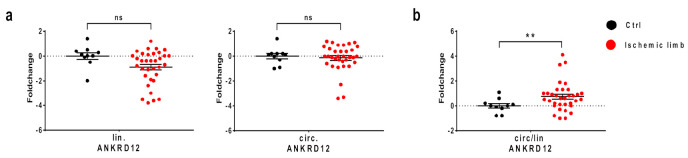
Increased ANKRD12 circular-to-linear ratio in ischemic muscles of CLI patients. (**a**) Expression levels of linear and circular ANKRD12 RNAs were measured by qPCR in skeletal muscle samples taken from the ischemic limb of CLI patients (*n* = 34) and compared to non-ischemic muscles of control subjects (*n* = 10). (**b**) circular-to-linear ratios (circ/lin) were determined. Fold changes are in log2 scale (−ΔΔCT) and individual values are indicated by dots, while mean and SEM are indicated by horizontal lines (** *p* < 0.01; ns: not significant).

**Figure 6 cells-11-03546-f006:**
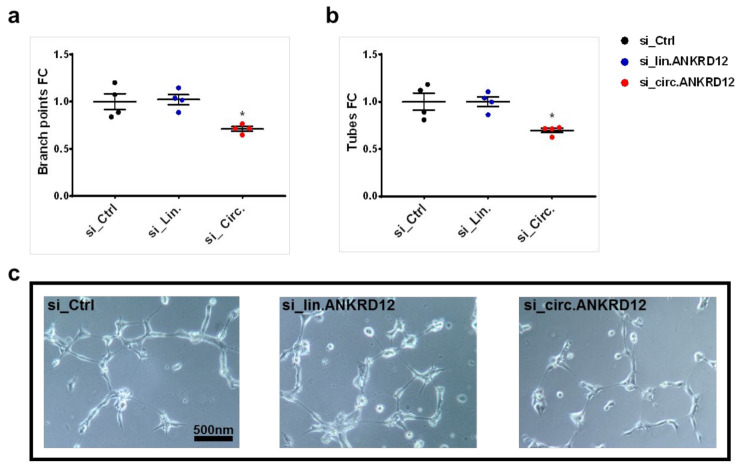
circANKRD12 knockdown decreases the formation of capillary-like structures. HUVEC were transfected with siRNAs specific for either the circular (si_Circ.) or the linear transcript (si_Lin.). As negative control, a non-targeting siRNA (si_Ctrl) was used. Forty-eight hours after transfection, the ability to form capillary-like structures by Matrigel assay was quantified. (**a**) Scatterplot of the number of branch points relative to control. (**b**) Scatterplot of the number of tubes relative to control. (**c**) Representative phase-contrast images of the organization into capillary-like structures for all transfections are shown. Individual values are indicated by dots, while mean and SEM are indicated by horizontal lines (*n* = 3, * *p* < 0.05).

**Figure 7 cells-11-03546-f007:**
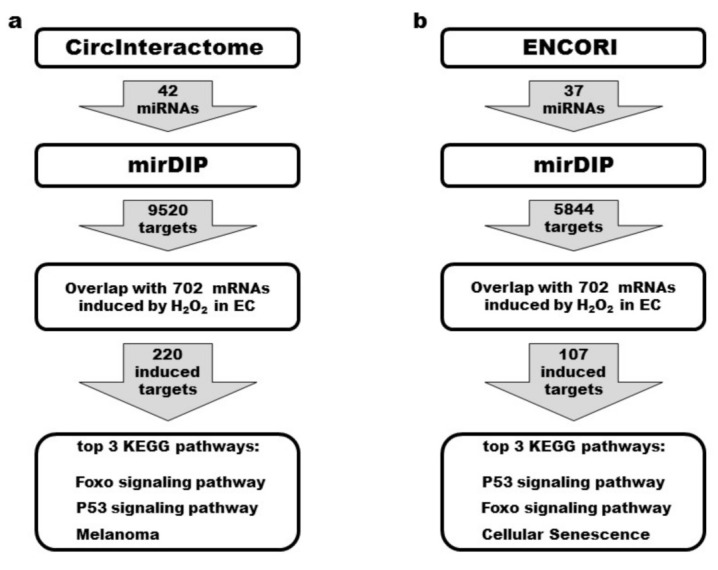
Pipelines for investigating circRNA–miRNA–mRNA networks of circANKRD12. Two different prediction tools identified miRNA binding sites for circANKRD12. (**a**) Circular RNA Interactome, based on the TargetScan prediction algorithm and (**b**) ENCORI, supported by Ago CLIP-seq experiments. Predicted targets identified by mirDIP were intersected with mRNAs found to be induced upon exposure to 16 or 36 h of H_2_O_2_ in HUVEC (GSE104664). KEGG-enriched pathways of the identified targets were ranked by FDR.

**Figure 8 cells-11-03546-f008:**
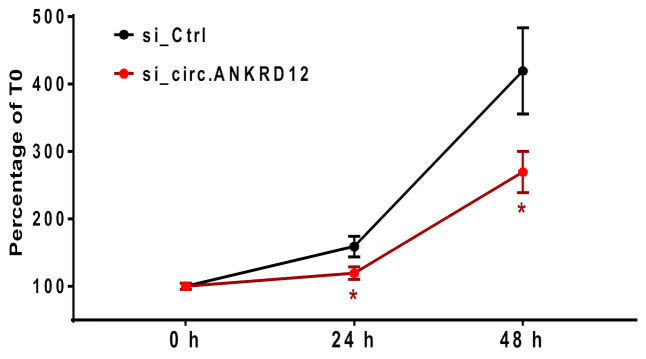
Knockdown of circular ANKRD12 leads to reduced endothelial proliferation. HUVEC were transfected with siRNA specific for the back-splice junction of circANKRD12. As a negative control a non-targeting siRNA (si_Ctrl) was used. Three independent proliferation assays (each *n* = 4) were aggregated by averaging cell numbers expressed as percentages relative to time point 0 (*n* = 12; * *p* < 0.05).

## Data Availability

The RNAseq data used in this study were previously published and deposited in the National Center for Biotechnology Information Gene Expression Omnibus database (Accession No. GSE104664).

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
