# Peer review of "CircANKRD12 Is Induced in Endothelial Cell Response to Oxidative Stress"

_cells, 2022, doi:10.3390/cells11223546_

Round 1
Reviewer 1 Report
The article by Voellenkle et al. studies the involvement of circRNAs in endothelial cells (ECs) exposed to oxidative stress. They found that circANKRD12 but no linANKRD12 plays an important role in ECs exposed to oxidative stress using H2O2 and hypoxia.
I believe this article would be a good addition to the already existing papers related to circRNAs and oxidative stress. However, some revisions are needed to be done before the paper can be published.
1. Please mention on what basis HUVECs were treated with H2O2 for 16 and 36 hours and 1% O2 for 24 and 48 hours.
2. Also please mention why you chose the concentration (50 nM) of transcript-specific ON-TARGETplus siRNAs.
3. Section 2.5, Line 134. HUVECs were seeded at a density of 18.000 cells per chamber or 18,000 cells per chamber?
4. Fig 2b, y-axis label is not clear.
5. Fig 6, please mention ANKRD12 for labeling the pictures and graphs
6. Fig 8, Could the authors provide some qualitative images?
7. Please mention full forms for FOXO, NRF2 etc. where it appears for the first time in the text.
Author Response
REVIEWER 1:
Thank you for your precise and detailed revision.
The whole manuscript was carefully proof-read.
For your convenience all changes in the manuscript are highlighted in yellow.
Q1: Please mention on what basis HUVECs were treated with H2O2 for 16 and 36 hours and 1% O2 for 24 and 48 hours.
A 1: In fact, this is a very important information, thank you for pointing it out.
These conditions were optimized and characterized in previous studies [Fuschi 2017, doi: 10.18632/aging.101341; Voellenkle 2012, doi: 10.1261/rna.027615.111; Voellenkle 2016 DOI: 10.1038/srep24141].
The H2O2 concentration used in this study induced a strong cytostatic effect and very low EC death in vitro. Therefore, the ensuing redox imbalance is likely close or below to levels expected to occur in many pathophysiological conditions, such as reperfusion injury, when oxidative stress can induce not only growth arrest but also extensive cell death [Mathru 1996, doi.org/10.1097/00000542-199601000-00003; Slezak, Am J Pathol. 1995; 147:772–81; Granger 2015, doi.org/10.1016/j.redox.2015.08.020].
The conditions used for the hypoxia experiments were characterized in detail in previous studies, showing the expected impact of this treatment on the coding and non-coding transcriptome. Amongst others, Enrichment analysis of Gene Ontology Biological Processes (GOBP) of genes modulated upon the described hypoxia exposure revealed significant involvements with categories related to hypoxia-response, angiogenic processes and cell cycle. (Fasanaro 2008, doi: 10.1074/jbc.M800731200, Voellenkle 2012, doi: 0.1261/rna.027615.111; Voellenkle 2016, doi: 10.1038/srep24141.)
Please find the relevant modifications in 2. Materials and Methods, section 2.2. page 3 lines 104-108
Q2: Also please mention why you chose the concentration (50 nM) of transcript-specific ON-TARGETplus siRNAs.
A 2: Thank you for bringing this missing information to our attention.
The chosen siRNA concentration was already optimized and used successfully for efficient knockdown in previous studies of our group and others (Guarani 2011, doi: 10.1038/nature09917; Helker 2020, doi: 10.7554/eLife.55589; Voellenkle 2016 doi: 10.1038/srep24141; Voellenkle 2019 doi: 10.3390/ijms20081938). Still, we verified the settings for the present investigation by preliminary experiments (data not shown).
Please find relevant modification in. 2. Materials and Methods, Section 2.2. page 3 lines 112-114
Q3: Section 2.5, Line 134. HUVECs were seeded at a density of 18.000 cells per chamber or 18,000 cells per chamber?
A 3: We apologize for this oversight. The correct number is 18 000.
We corrected accordingly in Section 2.5, page 3 line 143.
Q4: Fig 2b, y-axis label is not clear.
A 4: We apologize for this oversight.
Please find the modified Figure 2b in section 3.2, page 6 line 233-234
Q5: Fig 6, please mention ANKRD12 for labeling the pictures and graphs.
A 5: Thank you for this advice, it helps understanding on first sight.
Please find the modified Figure 6 in section 3.7, page 10 line 341-342
Q6: Fig 8, Could the authors provide some qualitative images?
A 6: Thank you for this recommendation.
Please find the images in the new Supplementary Figure 6
Q7: Please mention full forms for FOXO, NRF2 etc. where it appears for the first time in the text.
A7: Thank you for bringing this to our attention.
Please find full forms throughout the manuscript:
Abstract: page 1 line 22;
Introduction: page 2 lines 47, 48, 57-59, 77, 80;
Materials and Methods: page 3 lines 133, 133;
Results: page 6 lines 223, 224, 226, 227, page 11 line 376;
Discussion: page 12 lines 415-417, 419, 420

Reviewer 2 Report
Voellenkle et al aimed to decipher the molecular mechanisms involving circular RNAs (circRNAs) of the endothelial response to oxidative stress. In this context, they found that circANKRD12, derived from the junction of exon 2 and exon 8 of the ANKRD12 gene (hsa_circ_0000826), which was significantly induced in H2O2-treated ECs. Bioinformatics investigation of the circANKRD12-miRNA-mRNA regulatory network in H2O2-treated ECs determined the enrichment of the p53- and Foxo-signaling pathways. In keeping with the tumor suppressor role of the p53 pathway, circANKRD12 silencing repressed EC proliferation. In this way, the authors conclude that circANKRD12 is a significant player in EC exposed to oxidative stress.
Comments:
The article is well written.
The topic is original and relevant in the field.
I believe this study would be very useful for the clinical perspective of circANKRD12 in ECs.
I found the conclusion to be in line with the evidence and arguments presented.
The references are well-updated.
The figures are fine.
Some Suggestions to improve their manuscript:
Missing this study (PMID: 32987354) regarding p53 and its anti-proliferative role in response to DNA damage in cancer on Page no. 10, lines 373-374.
Include the latest review regarding p53 such as https://doi.org/10.3390/cancers14215176.
Nice Work!
Author Response
REVIEWER 2
Thank you for your precise and detailed revision.
The whole manuscript was carefully proof-read.
For your convenience all changes in the manuscript are highlighted in yellow.
Q1: Missing this study (PMID: 32987354) regarding p53 and its anti-proliferative role in response to DNA damage in cancer on Page no. 10, lines 373-374.
A 1: Thank you for bringing this very important article to our attention.
Please find it now as citation #55 in Results, page 11, line 391.
Q2: Include the latest review regarding p53 such as https://doi.org/10.3390/cancers14215176.
A 1: Thank you for bringing this very recent and interesting review to our attention.
Please find it now as citation #64 in Discussion, page 13, line 451, #64.
